# Non-Invasive Assessment of Abdominal/Diaphragmatic and Thoracic/Intercostal Spontaneous Breathing Contributions

**DOI:** 10.3390/s23249774

**Published:** 2023-12-12

**Authors:** Ella F. S. Guy, Jaimey A. Clifton, Jennifer L. Knopp, Lui R. Holder-Pearson, J. Geoffrey Chase

**Affiliations:** 1Centre for Bioengineering, Mechanical Engineering, University of Canterbury, Christchurch 8041, New Zealandjennifer.knopp@canterbury.ac.nz (J.L.K.); geoff.chase@canterbury.ac.nz (J.G.C.); 2Electrical and Computer Engineering, University of Canterbury, Christchurch 8041, New Zealand; lui.holder-pearson@canterbury.ac.nz

**Keywords:** circumference, monitoring, abdominal, thoracic, rotary encoder, respiratory mechanics, muscular recruitment

## Abstract

(1) Background: Technically, a simple, inexpensive, and non-invasive method of ascertaining volume changes in thoracic and abdominal cavities are required to expedite the development and validation of pulmonary mechanics models. Clinically, this measure enables the real-time monitoring of muscular recruitment patterns and breathing effort. Thus, it has the potential, for example, to help differentiate between respiratory disease and dysfunctional breathing, which otherwise can present with similar symptoms such as breath rate. Current automatic methods of measuring chest expansion are invasive, intrusive, and/or difficult to conduct in conjunction with pulmonary function testing (spontaneous breathing pressure and flow measurements). (2) Methods: A tape measure and rotary encoder band system developed by the authors was used to directly measure changes in thoracic and abdominal circumferences without the calibration required for analogous strain-gauge-based or image processing solutions. (3) Results: Using scaling factors from the literature allowed for the conversion of thoracic and abdominal motion to lung volume, combining motion measurements correlated to flow-based measured tidal volume (normalised by subject weight) with R^2^ = 0.79 in data from 29 healthy adult subjects during panting, normal, and deep breathing at 0 cmH_2_O (ZEEP), 4 cmH_2_O, and 8 cmH_2_O PEEP (positive end-expiratory pressure). However, the correlation for individual subjects is substantially higher, indicating size and other physiological differences should be accounted for in scaling. The pattern of abdominal and chest expansion was captured, allowing for the analysis of muscular recruitment patterns over different breathing modes and the differentiation of active and passive modes. (4) Conclusions: The method and measuring device(s) enable the validation of patient-specific lung mechanics models and accurately elucidate diaphragmatic-driven volume changes due to intercostal/chest-wall muscular recruitment and elastic recoil.

## 1. Introduction

Globally, the burden of respiratory disease to healthcare systems is increasing [1,2,3,4,5]. Currently, testing in home and primary care settings is limited. Most diagnoses and monitoring require specialised tests, such as spirometry, polysomnography, cardiopulmonary exercise testing, and full body plethysmography [6,7,8,9]. Hence, such testing is predominantly conducted in respiratory clinics or hospitals with trained personnel, which limit access and utility. Shifting screening and testing to home, community, and primary care settings would decrease the burden of respiratory testing on healthcare systems. Low-cost monitoring devices for these settings would need to collect comparable data to clinical testing with comparable accuracy, as well as provide and assess clinically examinable/observable and useful information.

Respiratory disease can be categorised by a combination of obstructive and/or restrictive anatomical abnormalities [7,9,10]. Obstructive abnormalities increase airway resistance by the occlusion of airways [9,11]. Occlusion can be caused by inflammation, such as in diseases like asthma, a lack of muscular airway support, such as in sleep disordered breathing, and mucus accumulation [9,11,12]. Restrictive abnormalities increase lung elastance, or stiffness, via tissue stiffening, often caused by lung scarring (fibrosis) from infection or inhaled particulates [9,13,14]. Extra-thoracic restrictive abnormalities, which can occur in obesity and burn victims, can also impact lung elastance [15,16]. Hence, observing and differentiating these disease states and underlying lung mechanics values or dynamics is important in diagnosis and management.

Primary clinical indicators of respiratory disease include observed breath patterns, palpation, percussion, auscultation and signs of under-oxygenation [9,17,18,19]. Normal breathing (eupnea) is typically 12–20 breaths per minute (bpm) for adults [9,20]. However, the ideal respiratory rate is patient-specific, and as a result, assessment methods and clinically defined acceptable ranges differ in clinical practice and publications [21]. An increased breath rate (tachypnea) can be an indicator of restrictive lung disease, differentiated by increased resistance to airflow [9,18,20]. However, rapid breathing with increased volume (hyperventilation) can be induced by increased metabolic demand in exercise (hypernea) [9]. Finally, breathing at a decreased rate is termed bradypnea or hypoventilation, when it also occurs at a decreased depth, and causes insufficient oxygenation [9,17,20].

A particular problem in diagnostic and management is respiratory disease caused by a physiological abnormality (obstructive, restrictive, or neuro-muscular), as it can be difficult to differentiate from purely dysfunctional breathing [22,23,24,25]. Dysfunctional breathing results in disordered breathing patterns, particularly excessive thoracic to abdominal/diaphragmatic muscular work [25,26,27,28]. In particular, identified dysfunctional breathing can be assessed and treated with appropriate physiotherapy [28,29,30,31], whereas treatment for respiratory disease often also includes pharmaceutical therapy, as well [8,32,33,34,35,36]. For example, respiratory muscle strength training is also beneficial in respiratory disease to maintain and optimise function [31], again, in addition to pharmaceutical therapy.

Clinically, disordered and dysfunctional breathing can be readily assessed [9,26,31,37,38,39,40]. Chest palpation allows for the assessment of the symmetry of chest expansion and the identification of regional abnormalities, such as, for example, lung collapse (pneumothorax) [9]. A Palpation and visual examination are also used to determine thoracic and abdominal breathing modes and ratios [28,31,40]. Percussion is used to determine the presence of fluid or material in the lung that may consolidate in the alveoli [9]. Auscultation, the assessment of breathing noises in the chest, allows clinicians to hear crackles from fibrosis in restrictive diseases and wheezes in obstructive diseases triggered bronchospasms [9,37]. Equally, while pulse oximetry is widely available and low-cost and can be used as an indirect measurement of respiratory function [38,39], its response is delayed, and it does not isolate the location or type of abnormality present [38,39].

The treatment of acute or chronic respiratory disease often includes some form of mechanical ventilation (MV) [41]. Invasive mechanical ventilation (IMV) requires sedation and intubation, with associated critical care, and so is limited to clinical settings [41,42,43]. Non-invasive mechanical ventilation (NIMV) is a common at-home therapy for sleep-disordered breathing, including obstructive sleep apnoea (OSA) [44,45,46,47].

Home-based monitoring requires an understanding of clinical indicators of respiratory abnormalities, distress, and patient–ventilator interaction. Models of pulmonary physiology have successfully identified lung elastance and resistance, with and without mechanical ventilation intervention [48,49,50,51,52,53,54]. Patient-specific drive has also been assessed using model-based methods [48,50,55,56]. The elucidation of muscular contributions and distribution of lung volume has now become a respiratory research focus [57,58,59,60,61,62]. However, to improve diagnosis, monitoring, and management, simple, non-invasive, and automated methods of assessing muscular recruitment and lung volume are required for the reliable differentiation of the causes of disordered breathing, particularly for use outside of fully equipped clinical settings.

To address this need, simple dynamic circumference rotary encoder tape measures were designed [63]. Two tape measures, around the chest and abdomen, provide a simple, quick, and very low-cost digital measure of thoraco-abdominal distribution of muscular recruitment and lung volume. These devices were designed to be easily integrated into current respiratory research data collection devices and are published and open access [63]. Preliminary trials presented here use dynamic circumference measures to provide a more complete, model-based picture of respiratory health, suitable for home-community diagnosis and monitoring.

## 2. Materials and Methods

Testing was conducted on 30 healthy adult subjects breathing normally, deeply (larger volumes at decreased rate), and panting (increased rate) at PEEP levels of 0 (ZEEP), 4, and 8 cmH_2_O [64,65]. One subject was excluded from this analysis due to an inspiratory flow sensor error in the 4 cmH_2_O deep breathing trial. Raw and processed datasets and descriptions are available and open access [64,65].

Pressure and flow were measured using differential pressure sensors over a 3D-printed venturi device, connected in series with CPAP circuitry at the airway opening (oral/nasal orifice) [64,65]. Dynamic thoracic and abdominal circumference was monitored using rotary encoder-based tape measure devices [63]. The devices consist of an inelastic fiberglass band spooled onto a barrel, which is connected to a rotary encoder [63]. The rotary encoder counts are used to translate barrel motion into linear change in unspooled tape [63]. The initial and any subsequent change in circumference as a function of time are captured as the patient breathes [63]. To capture chest motion, bands were placed at armpit level where the tape barrel sat on the flat upper part of the subject’s sternum. A second band was placed at the subject’s waist (narrowest point of lower trunk) to capture abdominal motion. Overall, the dataset provides a full set of respiratory measurements for the comparison and validation of these new sensors relative to clinical expectations and measurements.

The volume change measured by tapes was compared to tidal volume, measured as the integral of flow. Tape- and flow-based volume were compared to evaluate the two-point (thoracic and abdominal) tape method of ascertaining lung volume. Scaling factors for abdominal and diaphragmatic expansions to the recruited lung volume are available for subjects examined in a supine posture to compare to a surrogate of volume contribution [66]. Standing values were taken to best approximate the seated position used in this trial [66]:Scaled Expansion = 0.7 × Thoracic Expansion + 0.28 × Abdominal expansion,(1)

Subsequently, the abdominal and thoracic contributions were compared individually to elucidate their contributions to the tested PEEP levels and breathing types. The relative timing of thoracic and abdominal contributions were also compared to assess trends in muscular recruitment and manoeuvres within breaths, ultimately classifying breathing modes in the context of muscular recruitment.

Table 1 outlines demographic data, where self-reported data on asthma and smoking/vaping history are included in Table 2 and Table 3:

## 3. Results

Peak abdominal and chest expansions show a high degree of variation compared to tidal volumes over all breaths from all subjects and trials (Figure 1). Tidal volume is more highly correlated with scaled peak abdominal and/or chest motion (Figure 2). The scaled sum of thoracic and abdominal motion, measured by the rotary encoder tape measures, is compared to the tidal volume normalised by subject weight, as shown in Figure 3 (R^2^ = 0.79). The R^2^ value for scaled summed tape motion compared to tidal volume without weight normalisation is R^2^ = 0.72 (Figure 2). The unscaled summed tape motion compared to the tidal volume with and without weight normalisation is R^2^ = 0.75 and 0.69, respectively.

The scaled sum of chest and abdominal expansion against tidal volume normalised by subject weight is compared by demographics in Figure 4. Women show slightly less variability compared to men (Figure 4a). The two asthmatics show subject-specific trends but a large variation in comparison to each other (Figure 4b). Subject-specific trends can be similarly seen in vaper and smoker cohorts, albeit less clearly (Figure 4c,d). However, vapers are more highly correlated with lower-scaled expansions in relation to tidal volume normalised by weight (Figure 4c). Figure 5 illustrates the scaled sum of expansions over all breath types against tidal volume normalised by subject weight in subplots by subject to more clearly demonstrate the subject specificity of this trend.

Figure 6, Figure 7 and Figure 8 illustrate the chest and abdominal expansions for normal breathing, deep breathing, and panting tests, respectively. The trial subjects had a varied interpretation for panting and deep breathing (Figure 7 and Figure 8). In particular, in panting trials, some subjects exhibited abdominal-dominant contributions, and others, thoracic-dominant motion (Figure 8). However, in all tests, the abdominal–thoracic pattern was similar for each breath for a given subject (Figure 6, Figure 7 and Figure 8).

For each trial, the mean tidal volumes and circumferential changes were computed. Figure 9, Figure 10 and Figure 11 illustrate these mean breaths for Subject 3 at ZEEP, 4 cmH_2_O, and 8 cmH_2_O, respectively. Mean breath abdominal-to-chest circumference expansions were then plotted against each other in Figure 12 for Subject 3. As expected, circumferential changes are highest where tidal volume is largest (“deep breathing” in Figure 9, Figure 10 and Figure 11) and very small under low tidal volume and high respiration rate (“panting” in Figure 9, Figure 10 and Figure 11) across all PEEPs. The abdominal and chest circumference peaks are similar during normal and deep breathing, but the chest lags the abdomen when panting (Figure 9, Figure 10 and Figure 11). This pattern is typical in subjects during rapid shallow panting (Figure 8).

Figure 12 also illustrates the inspiratory–expiratory muscular shift in active (panting and deep) breathing compared to resting normal breathing. This can be seen as a wider loop in Figure 12, particularly in panting. Thus, this could be indicative of a volume redistribution from abdominally recruited lung volume to thoracically recruited volume in forced expiration.

Ratios of abdominal-to-chest motion for all breaths, and with increasing PEEP, are illustrated for each of the different breath types in Figure 13, Figure 14 and Figure 15. Figure 13 shows a slight increase in chest contributions with increased PEEP in normal breathing. There is an increased chest contribution in deep breathing (Figure 14) compared to normal breathing (Figure 13), indicative of the intercostal muscular requirement to recruit larger volumes. In panting, contributions show a wide distribution; however, it has a higher median abdominal contribution (Figure 15). This could suggest abdominal muscles can be more quickly recruited and thus more dominant in breathing at an increased rate, as can also be seen in Figure 9, Figure 10 and Figure 11.

## 4. Discussion

Individually, peak abdominal and chest expansions do not show a strong correlation to tidal volume over all breaths from all subjects and trials (Figure 1). Thus, a one-point tape system placed either abdominally or thoracically would not provide reliable tidal volume indication. Additionally, variations in abdominal and chest expansions relative to tidal volume show high subject specificity and breath-rate dependence (Figure 1, Figure 6, Figure 7 and Figure 8). Subject specificity and thus inter-subject variability in the results are likely due to a combination of physiological differences and variation in band placement and could be better compared to a measurement of height between tapes in future testing.

The scaled sum of thoracic and abdominal motion more highly correlates with tidal volume, as R2 = 0.79 (Figure 3). The degree of correlation of tape motion to flow-based tidal volume validates the two-tape circumference measurement method of the rotary encoder dynamic in estimating tidal volume over a variety of breathing modes. However, these correlations are still fairly variable (Figure 2 and Figure 3) and could be improved by an investigation into the subject specificity of these trends, as seen in Figure 5. There is likely a size dependency of this factor, indicated by the improved correlation when normalised by weight (Figure 3). Lung and thoracic elastance should also be included in modelling tidal volume from expansion, which is further indicated by Figure 4c, which shows a trend across vapers, who can be expected to have impacted lung elastance.

An offset is present in the comparison of tidal volume and tape motion of approximately 1 mL/kg subject weight (Figure 3). This offset is likely a combination of initial lung expansion during diaphragm dissention, causing abdominal pressurisation prior to expansion and skin tissue compression before tape motion. Hence, this offset could partially represents a “free volume” in lung expansion before chest and abdominal distension. Thus, theoretical panting at a tidal volume of 1–2 mL may result in no chest or abdominal movement.

The initial elucidation of abdominal and thoracic modes of breathing in normal breathing show that abdominal motion tends to be greater than chest motion (Figure 6, Figure 7, Figure 8, Figure 9, Figure 10 and Figure 11). This difference was expected, as the primary muscle of respiration is the diaphragm in normal breathing (without dysfunction). With increasing PEEP, the ratio of abdominal to chest motion in normal breathing diminishes (Figure 13, Figure 14 and Figure 15). This change could be due to the controlled distention of the diaphragm against the PEEP expansion pressure, an inefficient pattern in response to the sensation of applied PEEP, or an indicator the PEEP causing diaphragmatic distension prior to inhalation, which is then not captured in the resulting dynamic breath motion.

Both deep and normal breathing showed relatively synchronised thoraco-abdominal motion (Figure 6, Figure 7, Figure 9 and Figure 10), indicative of uniform muscular recruitment. In deep breathing, chest motion was increased (Figure 7 and Figure 10) to facilitate the larger tidal volumes. As PEEP is increased, an increased variation in the ratio of abdominal-to-chest motion was observed (Figure 14), although median motion ratios remained relatively consistent between PEEP levels. Muscular recruitment and breathing pattern are particularly important in sports, singing, and the management of chronic diseases [67,68,69]. Thus, feedback on the recruitment and distribution of lung volume has potential application in respiratory physiotherapy and the monitoring of diseases.

Panting showed a high degree of variation for relatively small tidal volumes and fast breath rates (Figure 8 and Figure 11). Some variation could be accounted for by the subject-specific interpretation of panting as either short active inhalations, exhalations, or both (Figure 8). In panting, the abdomen typically leads the chest in inspiration (Figure 8 and Figure 11), indicating the diaphragm can be more quickly actively recruited. The diaphragm also leads the chest in expiration in some cases of panting (Figure 8 and Figure 11), indicative of forced abdominal expiration, where the volume is redistributed from the abdominal to the thoracic region at the onset of expiration. The rapid recruitment of the diaphragm would also explain the increase in abdominal-to-chest expansion seen in panting (Figure 15) to accommodate an increased respiratory rate.

This abdominal–thoracic redistribution can be more clearly seen in the loops in Figure 12, while in normal breathing, inspiration and expiration follow a similar abdominal-to-thoracic expansion profile (Figure 12). Deep breathing shows an increased chest contribution in expiration, resulting in a looped profile (Figure 12), and the highest degree of looping is seen in panting (Figure 12). Hence, this shows the potential to differentiate active from passive breathing by the difference in thoracic to abdominal expansion ratio between inspiration and expiration.

Dynamic circumference monitoring using the rotary encoder tape measures from this trial does not provide an indication of lung volume distribution between the right and left lung. Future iterations could fix tapes at the spine and sternum to assess the symmetry of ventilation. In addition, future works could reassess tape placement, separating upper and lower abdominal distensions and calibrating the placement by rib count or something similar in order to improve the consistency of tape placement and ensure the abdominal tapes are not affected by lower rib movement. Structured light methods could also be used in a research setting [62], to give greater detail for overall chest movement. However, this study shows that a simple tape-based study of breathing volume changes can observe key features of tidal breathing, which may be useful for at-home or clinic-based assessments or monitoring.

Studying the impact of PEEP on functional residual capacity (FRC) can improve the understanding of the initial impact of PEEP on required muscular effort and lung contributions. The lungs would be expected to be have more lung volume at a higher PEEP, which is the function of PAP in preventing airway closure. However, in conscious healthy subjects, auto-PEEP would be a factor to consider, and it can also occur in intensive care unit patients, as well [70].

Subsequent trials could take observed breath patterns from these un-cued trials and direct patients to perform these breath types more precisely at cued rates and/or with the clinical guidance of breathing patterns. Breath patterns could include active inspiratory, expiratory, combined panting, and diaphragmatic breathing. Deep breathing at a cued rate, double breathing, and multiple breath patterns could also be considered. The aim of such trials would be to better understand the work of breathing under the different respiration rates seen across a broad range of disease conditions.

Additional testing on subjects with diagnosed respiratory disease and/or healthy patients with simulated respiratory disease would provide data on the pathological implications of muscular recruitment and breath patterns, thus informing predictive models of respiratory state and responses to PAP therapy and associated PEEP settings. Tests could be conducted before and after respiratory physiotherapy, which would provide an indication of breathing mode progression/modification. Testing could also be conducted alongside other aeration assessment techniques to compare accuracy.

## 5. Conclusions

Abdominal and thoracic dynamic circumference can be easily measured with sufficient accuracy using rotary encoder tape measures. The addition of a subject-specific scaling factor or model would likely enable lung volume to be predicted accurately from a two-tape system (at thoracic and abdominal locations). The measurement of the relative thoracic and abdominal contributions to breathing allows for the characterisation of breath types and patterns at different rates. With the potential to differentiate active and passive breathing modes. Hence, the data and methods have a potential application in the development of remote respiratory assessment tools, particularly at a low cost for at-home monitoring and guidance of care.

## Figures and Tables

**Figure 1 sensors-23-09774-f001:**
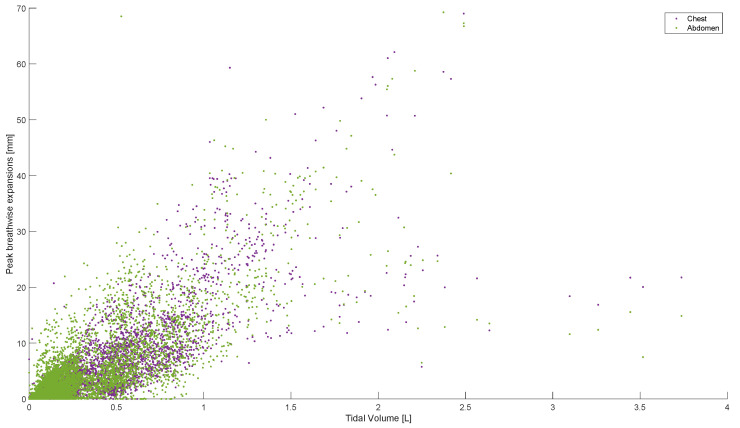
Breath-wise peak chest and abdominal circumferential changes (mm) against tidal volume (L). For men and women breathing normally, deeply, and panting at ZEEP, 4, and 8 cmH_2_O CPAP-set PEEP.

**Figure 2 sensors-23-09774-f002:**
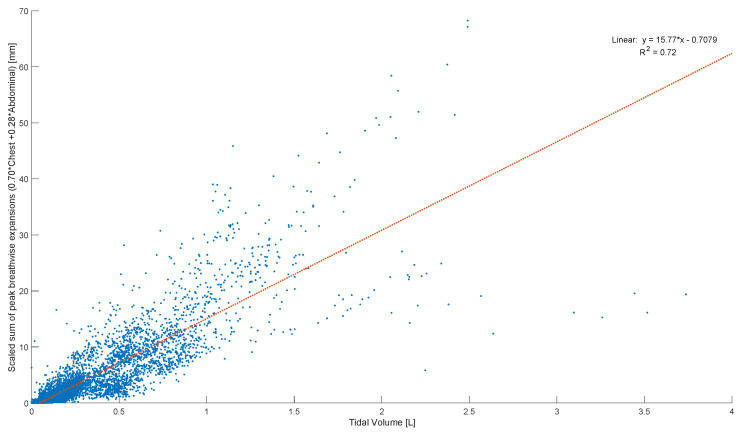
Breath-wise scaled sum of peak circumferential changes in chest and abdomen (mm) against tidal volume normalised by subject weight (L/kg). For men and women breathing normally, deeply, and panting at ZEEP, 4, and 8 cmH_2_O CPAP-set PEEP.

**Figure 3 sensors-23-09774-f003:**
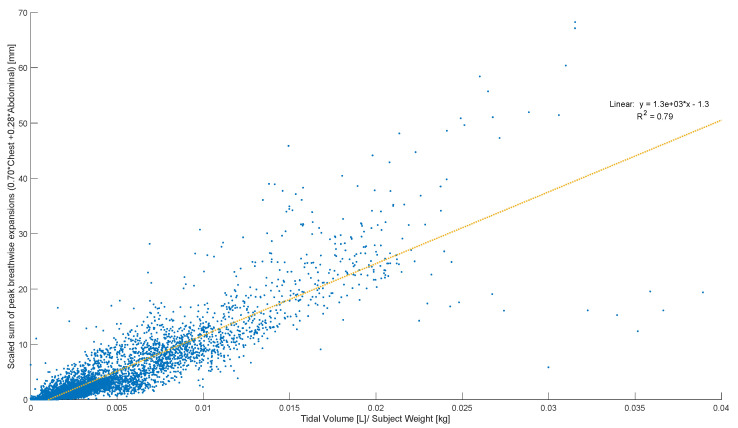
Breath-wise scaled sum of peak circumferential changes in thorax and abdomen (mm) against tidal volume normalised by subject weight (L/kg). For men and women breathing normally, deeply, and panting at ZEEP, 4, and 8 cmH_2_O CPAP-set PEEP.

**Figure 4 sensors-23-09774-f004:**
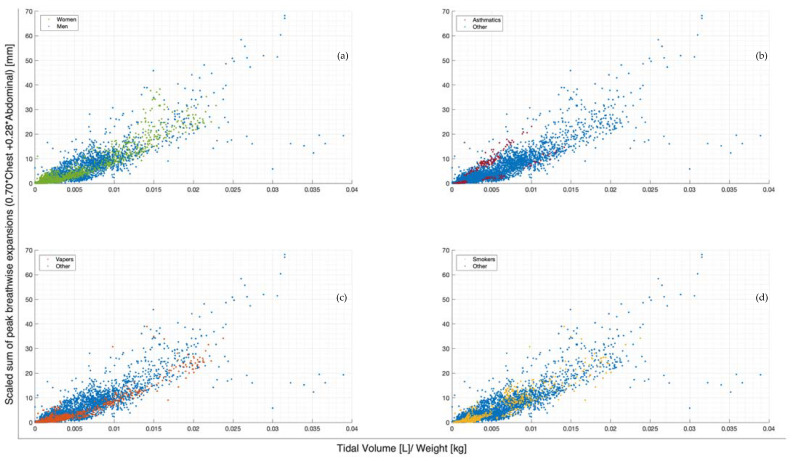
Breath-wise scaled sum of peak circumferential changes in thorax and abdomen (mm) against tidal volume normalised by subject weight (L/kg). For subjects breathing normally, deeply, and panting at ZEEP, 4, and 8 cmH_2_O CPAP-set PEEP. Separated by men and women (**a**), asthmatics and non-asthmatics (**b**), vapers and non-vapers (**c**), and smokers and non-smokers (**d**).

**Figure 5 sensors-23-09774-f005:**
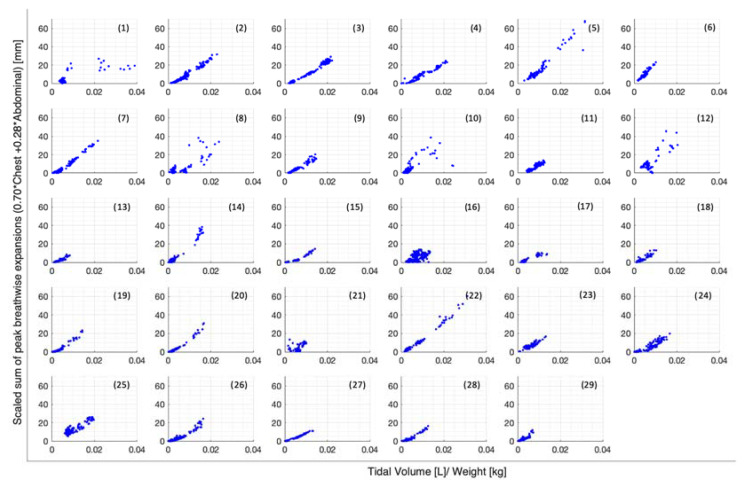
Breath-wise scaled sum of peak circumferential changes in thorax and abdomen (mm) against tidal volume normalised by subject weight (L/kg), separated by subject.

**Figure 6 sensors-23-09774-f006:**
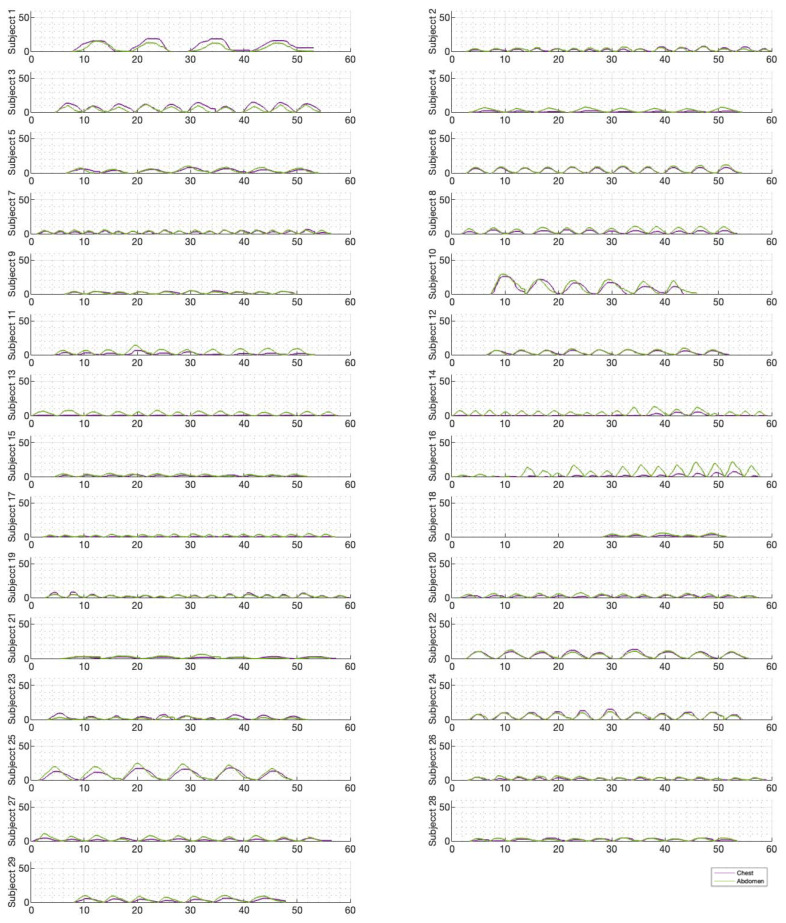
Chest and abdominal expansions (mm) over time (s) during normal breathing at ZEEP by subject.

**Figure 7 sensors-23-09774-f007:**
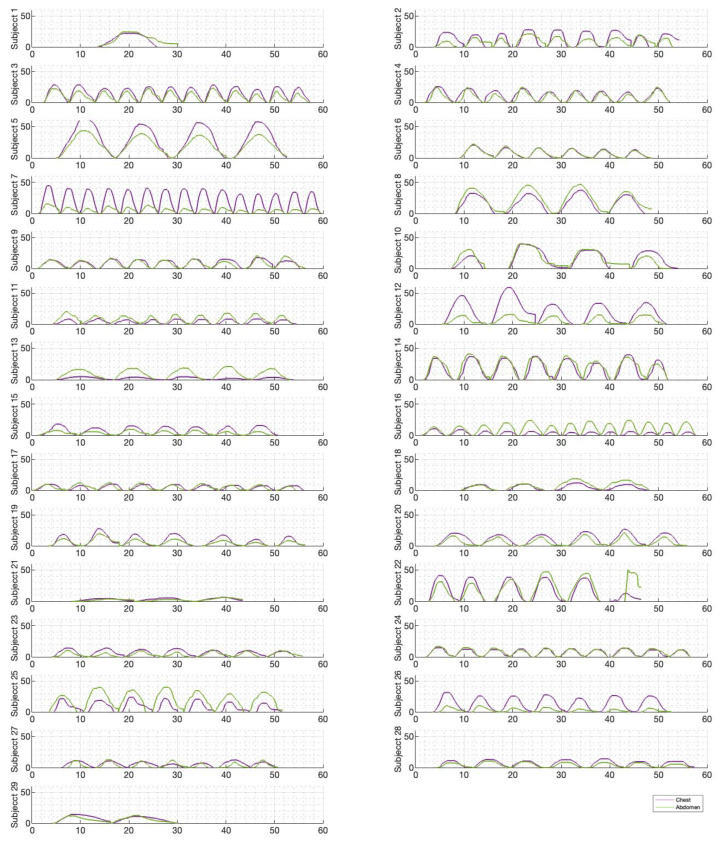
Chest and abdominal expansions (mm) over time (s) during deep breathing at ZEEP by subject.

**Figure 8 sensors-23-09774-f008:**
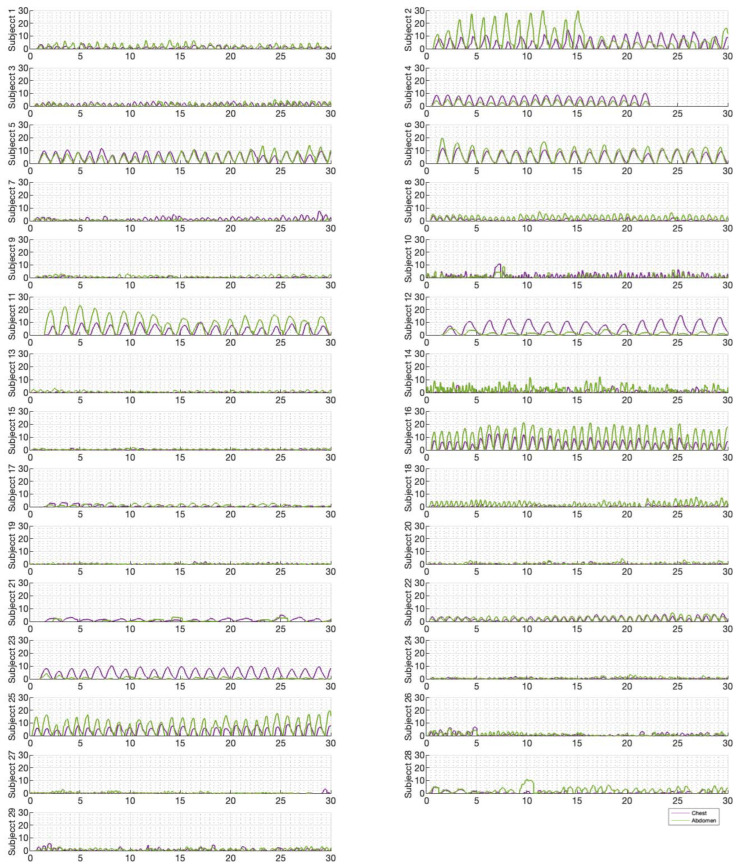
Chest and abdominal expansions (mm) over time (s) during panting at ZEEP by subject.

**Figure 9 sensors-23-09774-f009:**
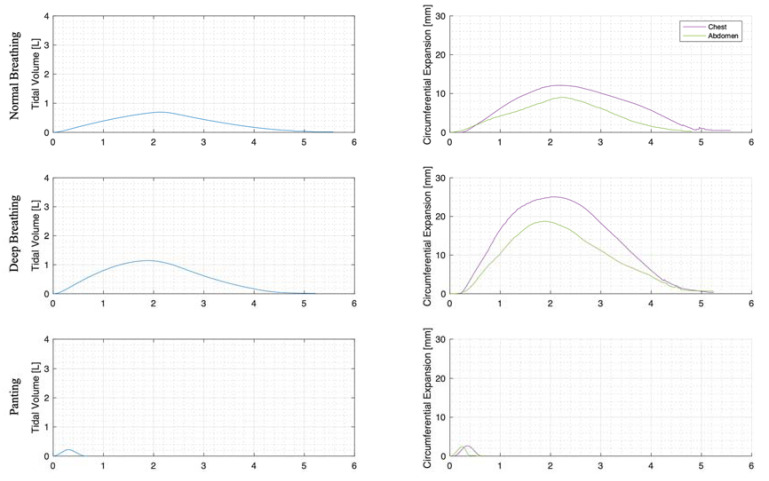
Subject 3 mean normal, deep, and panting breaths at ZEEP (against time (s)).

**Figure 10 sensors-23-09774-f010:**
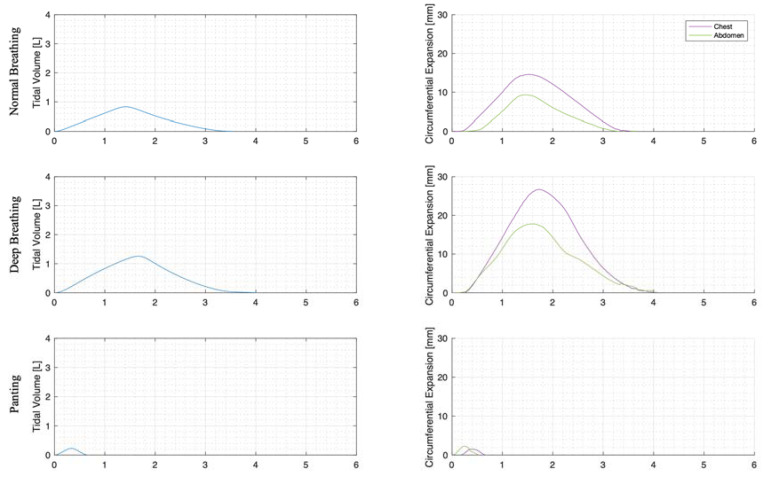
Subject 3 mean normal, deep, and panting breaths at 4 cmH_2_O (against time (s)).

**Figure 11 sensors-23-09774-f011:**
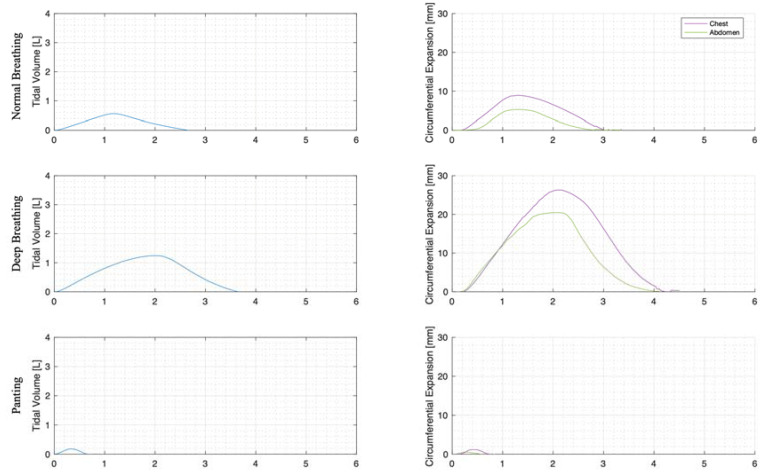
Subject 3 mean normal, deep, and panting breaths at 8 cmH_2_O (against time (s)).

**Figure 12 sensors-23-09774-f012:**
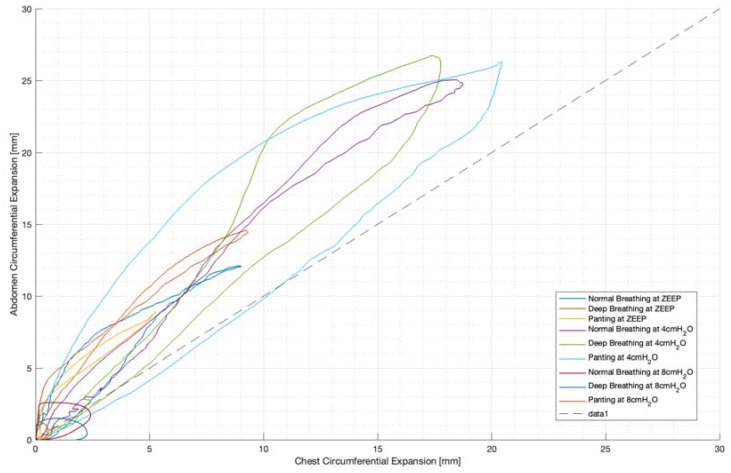
Subject 3 mean breath chest vs. abdominal expansions (mm).

**Figure 13 sensors-23-09774-f013:**
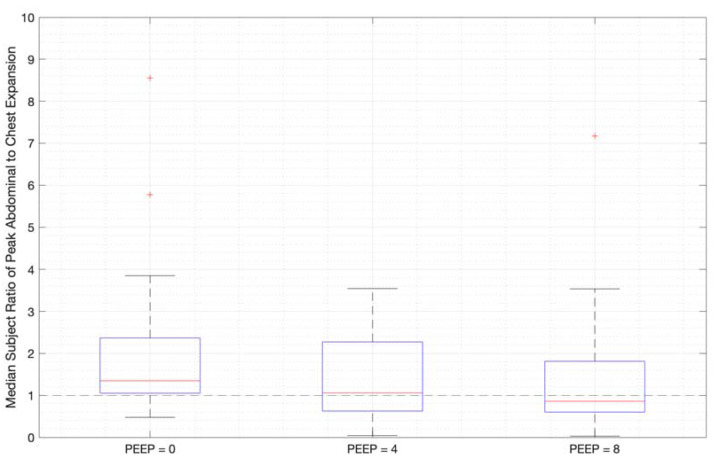
Median subject ratios of peak abdominal-to-chest motion over PEEP for normal breathing (outliers above 10 times abdominal-to-chest peak expansion are not shown for clarity).

**Figure 14 sensors-23-09774-f014:**
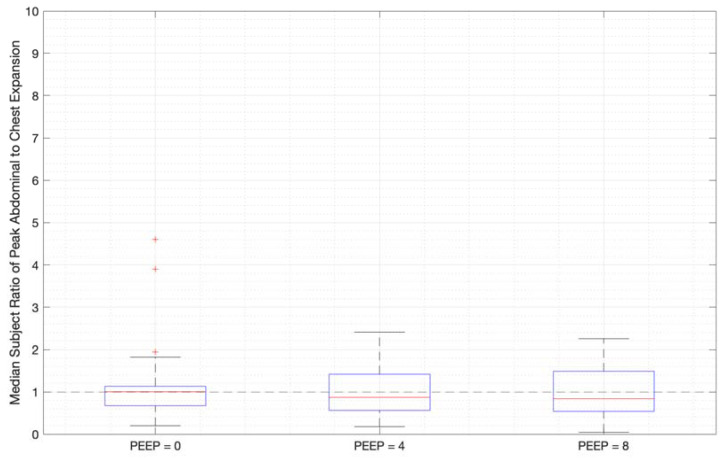
Median subject ratios of peak abdominal-to-chest motion over PEEP for deep breathing (outliers above 10 times abdominal-to-chest peak expansion are not shown for clarity).

**Figure 15 sensors-23-09774-f015:**
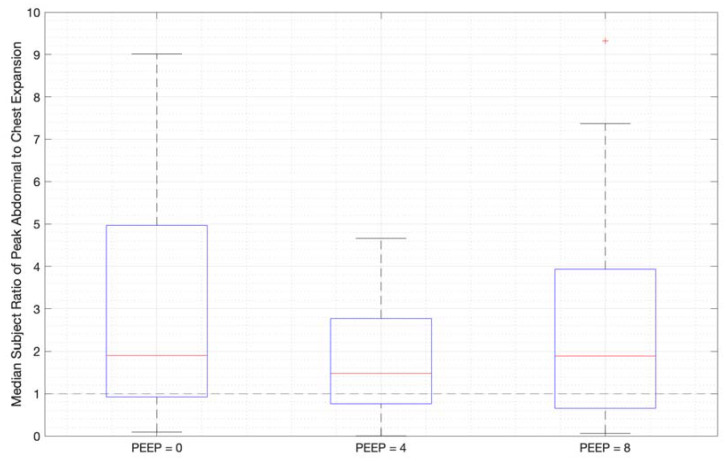
Median subject ratios of peak abdominal-to-chest motion over PEEP -for panting (outliers above 10 times abdominal-to-chest peak expansion are not shown for clarity).

**Table 1 sensors-23-09774-t001:** Subject demographic data.

Subject	Sex (M/F)	Age (Years)	Weight (kg)	Height (cm)	BMI	Asthmatic (Y/N)	Smoker/Vaper (Y/N)
1	M	37	96	177	30.6	N	N
2	F	23	65	169	22.8	N	N
3	F	21	63	171	21.5	N	Y
4	M	23	105	183	31.4	N	N
5	M	26	79	178	24.9	N	N
6	M	23	65	183	19.4	Y	N
7	F	20	60	170	20.8	N	N
8	M	23	75	182	22.6	N	Y
9	F	21	60	165	22.0	N	N
10	M	23	75	172	25.4	N	N
11	M	23	79	168	28.0	N	N
12	M	26	77	183	23.0	N	N
13	F	21	71.5	179	22.3	N	N
14	F	20	72	167	25.8	N	N
15	F	22	80	173	26.7	Y	Y
16	M	22	70	188	19.8	N	N
17	F	19	50	165	18.4	N	Y
18	F	27	73	153	31.2	N	N
19	F	23	57	158	22.8	N	Y
20	F	19	69	164	25.7	N	N
21	M	21	90	176	29.1	N	N
22	M	22	76.6	187	21.9	N	N
23	M	35	108	185	31.6	N	N
24	M	23	75	163	28.2	N	N
25	M	23	80	186	23.1	N	Y
26	F	20	55	163	20.7	N	N
27	F	22	97	169	34.0	N	Y
28	F	21	78	183	23.3	N	Y
29	F	20	52.8	160	20.6	N	N

**Table 2 sensors-23-09774-t002:** Asthmatic-specific subject demographic data.

Subject	Medication Used	Frequency of Use
6	Ventolin	<once weekly
15	Vanair (Budesonide + Formoterol)	2 times daily

**Table 3 sensors-23-09774-t003:** Smoker/Vaper-specific subject demographic data.

Subject	Frequency	Duration
3	<once daily	2 years
8	<once daily	2 years
15	10 inhales daily	1.5 years
17	150–200 inhales daily	2 years
19	Once a day	4 years
25	20 inhales daily	Intermittent
27	3 times a day	1 year
28	10 times a day	2 months

## Data Availability

Subject data are available and open access on PhysioNet and as a description in a research note article in Scientific Data [64,65].

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
