# Peer review of "Non-Invasive Assessment of Abdominal/Diaphragmatic and Thoracic/Intercostal Spontaneous Breathing Contributions"

_sensors, 2023, doi:10.3390/s23249774_

Round 1
Reviewer 1 Report
Comments and Suggestions for Authors
This paper provides a tape measure and rotary encoder band system developed by the authors was used to directly measure changes in thoracic and abdominal circumferences, and it have no need the calibration required for analogous strain-gauge based or image processing solutions. In addition, the research and experiments done by the author are detailed, but I would like to make the following suggestions here.
1.In the article, I am not able to know the physical information such as the shape of the measuring equipment, so please explain the direct picture of the shape of the measuring equipment in detail.
2.What is the formula in line 130 based on?
3.Does the equipment meet the sensitivity requirements, please support with data.
Author Response
Reviewer 1: Thank you for your time and feedback, your points have been individually addressed below.
- In the article, I am not able to know the physical information such as the shape of the measuring equipment, so please explain the direct picture of the shape of the measuring equipment in detail.
REPLY: Thank you, the physical information on measuring equipment is outlined in the referenced HardwareX paper, as for subject confidentially and ethics purposes we have no photos of the trial itself.
- What is the formula in line 130 based on?
REPLY: The reference to the scaling factor article as well as the purpose of this equation has been expanded on in the materials and methods section, as follows:
“Scaling factors for abdominal and diaphragmatic expansions to recruited lung volume are available for subjects in a supine posture to compare a surrogate of volume contribution [54]. Standing values were taken to best approximate the seated position used in this trial [54]:
Scaled Expansion = 0.7 * Thoracic Expansion + 0.28 *Abdominal expansion, (1)”
Reference [54] in the manuscript is:
“[54] Konno, K.; Mead, J. Measurement of the separate volume changes of rib cage and abdomen during breathing. Journal of applied physiology 1967, 22, 407-422”
- Does the equipment meet the sensitivity requirements, please support with data.
REPLY: These methods are not designed to be used solely as a diagnostic tool. Hence, it currently has no established specificity or sensitively requirements in diagnosis or diagnostic use. Rather, these methods, in pilot clinical testing with human subjects, have shown a research and potential therapeutic utility in the assessment of breathing mode without clinical training. This aspect has been expanded in the limitations/ future work section of the discussion to make this point explicit, as follows:
“Additional testing on subjects with diagnosed respiratory disease or/and healthy patients with simulated respiratory disease would provide data on the pathological implications to muscular recruitment and breath patterns, thus informing predictive models of respiratory state and responses to PAP therapy and associated PEEP settings. Tests could be conducted before and after respiratory physiotherapy, which would provide indication of breathing mode progression/modification. Testing could also be conducted alongside other aeration assessment techniques to compare accuracy.”
Reviewer 2 Report
Comments and Suggestions for Authors
The innovation of this manuscript is to study a simple, inexpensive, and non-invasive method of ascertaining volume changes in the thoracic and abdominal cavities. After careful reading of this manuscript, I have the following comments:
(1) The study have its novelty and the study is good design.
(2) The study is well conduct and data is good analysed and presented.
(3) The manuscript is good organized and written, with good conclusion
(4) As shown in Fig. 2 and 3, the linearity is just fair (r2 ranged 0.72~0.79). please try to describe and discuss more on this.
(5) A Bland–Altman plot must be conducted to prove this method.
Author Response
Reviewer 2: Thank you for your time and feedback, your points have been addressed below.
- The study have its novelty and the study is good design
- The study is well conduct and data is good analysed and presented.
- The manuscript is good organized and written, with good conclusion
REPLY: Thank you
- As shown in Fig. 2 and 3, the linearity is just fair (r2ranged 0.72~0.79). please try to describe and discuss more on this.
REPLY: Correct, in general, although in medicine 0.7-0.9 is considered “good” given the natural variation of and between subjects / patients. However, this area could use clarification, and so we have expanded in this discussion:
“Individually, peak abdominal and chest expansions do not show a strong correlation to tidal volume, over all breaths from all subjects and trials (Figure 1). Thus, a one-point tape system placed either abdominally or thoracically would not provide reliable tidal volume indication. Additionally, variation in abdominal and chest expansions relative to tidal volume show high subject specificity and breath-rate dependence (Figures 1, and 6-8). Subject-specificity and thus inter-subject variability in results is likely due to a combination of physiological differences and variation in band placement, and could be better compared with a measurement of height between tapes in future testing.
The scaled sum of thoracic and abdominal motion more highly correlates with tidal volume, R2 = 0.79 (Figure 3). The degree of correlation of tape motion to flow-based tidal volume validates the two-tape circumference measurement method of the rotary encoder dynamic in estimating tidal volume over a variety of breathing modes. However, these correlations are still fairly variable (Figures 2 and 3) which could be improved by investigation of the subject specificity of these trends, as seen in Figure 5. There is likely a size dependency of this factor, indicated by the improved correlation when normalised by weight (Figure 3). Lung and thoracic elastance should also be included in modelling tidal volume from expansion, which is further indicated by Figure 4c which shows a trend across vapers, who could be expected to have impacted lung elastance.”
The reviewer also asked: A Bland–Altman plot must be conducted to prove this method.
REPLY: A Bland-Altman plot was not conducted as we had no directly comparable gold-standard or reference measurement or metric to validate the bands against, which is required for a Bland-Altman plot. However, the extensions were validated to a gold standard in the referend HardwareX article [REFERENCE #51 in article]. We do however agree this approach would best done with bands in-situ and will consider this for future trials.
Reviewer 3 Report
Comments and Suggestions for Authors
Please see below a list of possible improvements for the paper:
1) For all the figures with the tidal volume, the number of points >2l tidal volume seems to be outliers, does the analysis changes if those are removed? Possibly a different linear interpolation should be considered for the two ranges??
2) Please make the lines on the plots thicker
3) does the model also works when there is respiratory effort measured as phase difference between upper and lower chest? if yes, to which extent?
Author Response
Reviewer 3: Thank you for your time and feedback, your points have been individually addressed below.
- For all the figures with the tidal volume, the number of points >2l tidal volume seems to be outliers, does the analysis changes if those are removed? Possibly a different linear interpolation should be considered for the two ranges??
REPLY: Yes, we agree to some extent. importantly, the data has not been removed in any plots as in Figure 5 the trends are indicated to be subject specific which accounts for the variation in the combined plots. Thus for accuracy and transparency all data has been included. However, we agree it needs clarification, and thus we have expended upon this point in new additions to the discussion, reading:
“Individually, peak abdominal and chest expansions do not show a strong correlation to tidal volume, over all breaths from all subjects and trials (Figure 1). Thus, a one-point tape system placed either abdominally or thoracically would not provide reliable tidal volume indication. Additionally, variation in abdominal and chest expansions relative to tidal volume show high subject specificity and breath-rate dependence (Figures 1, and 6-8). Subject-specificity and thus inter-subject variability in results is likely due to a combination of physiological differences and variation in band placement, and could be better compared with a measurement of height between tapes in future testing.
The scaled sum of thoracic and abdominal motion more highly correlates with tidal volume, R2 = 0.79 (Figure 3). The degree of correlation of tape motion to flow-based tidal volume validates the two-tape circumference measurement method of the rotary encoder dynamic in estimating tidal volume over a variety of breathing modes. However, these correlations are still fairly variable (Figures 2 and 3) which could be improved by investigation of the subject specificity of these trends, as seen in Figure 5. There is likely a size dependency of this factor, indicated by the improved correlation when normalised by weight (Figure 3). Lung and thoracic elastance should also be included in modelling tidal volume from expansion, which is further indicated by Figure 4c which shows a trend across vapers, who could be expected to have impacted lung elastance”
- Please make the lines on the plots thicker
REPLY: Thank you, these changes have been made.
- does the model also works when there is respiratory effort measured as phase difference between upper and lower chest? if yes, to which extent?
REPLY: Deep and panting breaths which were out of phase were include in trend analysis, and showed consistent subject-specific trends in Figure 5. This has been expanded upon in the discussion, as follows:
“The scaled sum of thoracic and abdominal motion more highly correlates with tidal volume, R2 = 0.79 (Figure 3). The degree of correlation of tape motion to flow-based tidal volume validates the two-tape circumference measurement method of the rotary encoder dynamic in estimating tidal volume over a variety of breathing modes. However, these correlations are still fairly variable (Figures 2 &3) which could be improved by investigation of the subject specificity of these trends, as seen in Figure 5. There is likely a size dependency of this factor, indicated by the improved correlation when normalised by weight (Figure 3). Lung and thoracic elastance should also be included in modelling tidal volume from expansion, which is further indicated by Figure 4c which shows a trend across vapers, who could be expected to have impacted lung elastance.”